# A Review of Degradation Models and Remaining Useful Life Prediction for Testing Design and Predictive Maintenance of Lithium-Ion Batteries

**DOI:** 10.3390/s24113382

**Published:** 2024-05-24

**Authors:** Gabriele Patrizi, Luca Martiri, Antonio Pievatolo, Alessandro Magrini, Giovanni Meccariello, Loredana Cristaldi, Nedka Dechkova Nikiforova

**Affiliations:** 1Department of Information Engineering, University of Florence, 50139 Florence, Italy; 2Department of Electronics, Information and Bioengineering, Polytechnic of Milan, 20133 Milan, Italy; luca.martiri@polimi.it (L.M.); loredana.cristaldi@polimi.it (L.C.); 3Institute for Applied Mathematics and Information Technologies “E. Magenes”, National Research Council, 20133 Milan, Italy; 4Department of Statistics, Computer Science, Applications “G. Parenti”, University of Florence, 50134 Florence, Italy; alessandro.magrini@unifi.it (A.M.); n.nikiforova@unifi.it (N.D.N.); 5Institute of Sciences and Technologies for Energy and Sustainable Mobility, National Research Council, 80125 Naples, Italy; giovanni.meccariello@stems.cnr.it

**Keywords:** degradation modelling, remaining useful life, accelerated degradation tests, predictive maintenance, reinforcement learning

## Abstract

We present a novel decision-making framework for accelerated degradation tests and predictive maintenance that exploits prior knowledge and experimental data on the system’s state. As a framework for sequential decision making in these areas, dynamic programming and reinforcement learning are considered, along with data-driven degradation learning when necessary. Furthermore, we illustrate both stochastic and machine learning degradation models, which are integrated in the framework, using data-driven methods. These methods are presented as a valuable tool for designing life-testing experiments and for maintaining lithium-ion batteries.

## 1. Introduction

As lithium-ion batteries have become increasingly common, estimating their remaining useful life (RUL) has become a necessity due to their impact on system availability and safety. RUL is especially useful for establishing predictive maintenance strategies due to its prognostic value. Battery degradation models should also incorporate the effects of different usages and environmental conditions on battery performance to make reliable predictions of the RUL. Battery degradation behavior must be characterized through accelerated degradation tests, which are planned based on optimal design theory to predict RULs and discriminate among competing models. An optimal maintenance strategy can be chosen by using reinforcement learning methods based on a good degradation model. A brief overview of all these methods is presented in this article. Individually, they are well-represented in the literature, but considering them together is a novel approach to maintenance. Due to the fact that batteries are often used in uncontrolled environments, the combined policy and model learning aspects of this approach seem particularly promising.

A lithium-ion battery’s state of health (SoH) decays exponentially during the degradation process. Various methods can be used to estimate the SoH parameter, including direct estimation from the discharge capacity or open circuit voltage (OCV), sensor fusion algorithms, or indirect processing from equivalent series resistance (ESR). Several factors contribute to the degradation of batteries, including battery chemistry, size, and operating conditions. It is important to note, however, that the general trend is always characterized by an exponential decay, as shown in Figure 1 and Figure 2 (see [1,2]). A detailed description of the possible approaches to degradation modelling is provided in Section 2, followed by a description of specific models that can be used to represent the exponential decay of SOH in Section 3. Section 4 discusses the design of experiments for accelerated testing (i.e., optimal designs), as well as reinforcement learning for maintenance planning. The last section, Section 5, of the paper is a comment section.

## 2. Approaches to Degradation Modelling

Within the context of prognostic and health management (PHM) fall several different techniques used to analyze the degradation processes in lithium-ion batteries. The final aim is usually the prediction of their remaining useful life (RUL). The literature presents two distinct classes of approaches for this purpose, as detailed in [3,4].

The first category is represented by physics of failure (PoF) models that are used in prognostics and remaining useful life (RUL) estimation to understand the underlying physical mechanisms that lead to the degradation and failure of a system over time. These models are based on the fundamental principles of physics and engineering to predict how various stresses and environmental factors influence the health and performance of a component or system. For this reason, PoF models are not commonly employed in the case of energy storage systems due to the complex non-linear degradation mechanisms that dominate the chemical wear-out of batteries [5,6].The second category is called data-driven methods because they rely on the analysis of historical or real-time data to predict the future health and the degradation path of a system. In order to do that, data-driven methods leverage patterns and information directly obtained from the system’s operational data as well as from the environmental conditions (which are usually called covariates). Typically, data-driven models are based on a multi-step procedure, starting from the data collection (either from the actual system or from historical datasets) followed by a feature extraction phase, a preprocessing phase (like anomaly detection, patter recognition, clustering, regression, and so on), and a training phase. After that, the model needs to be validated before it can be actually applied online on the system to estimate the RUL (eventually associated with a confidence interval or an uncertainty assessment). In the context of lithium-ion batteries, common data-driven models can be divided in the following sub-categories:–Stochastic models based on probabilistic assumptions, which include general path models [7], and stochastic processes like autoregressive integrated moving average (ARIMA) models [8], the Wiener process [9], the Brownian motion process [10], the gamma process [11], and the inverse Gaussian process [12].–Recursive mathematical filtering algorithm like the Kalman filter [13] and the particle filter [14], as well as their improvements and extensions. These methods are used to dynamically estimate the state of a battery, taking into account both the measurements and the predicted system state.–The alternative is represented by machine learning (ML) models. ML is a subset of artificial intelligence that enables systems to learn and make predictions or decisions without being explicitly programmed. It involves designing and developing algorithms and models that can learn patterns and relationships from data and use them to make predictions or take actions. ML algorithms are designed to improve their performance over time through experience, adjusting and optimizing their models based on feedback and new data. In the case of lithium batteries, common ML algorithms for PHM and RUL prediction include but are not limited to: support vector machine [15], relevance vector machine [16], random forest regression [17], artificial neural network [18], variational autoencoders [19], and deep neural networks. Examples of the latter are long short-term memory network (LSTM) [20], temporal transformer network [21], deep neural network [22], and echo state network [23]. An overall review of ML techniques for RUL estimation of batteries in recent years is presented in [24].

The recent literature is increasingly favoring data-driven approaches due to challenges in modelling battery degradation trends. Data-driven models offer enhanced accuracy of RUL prediction with a consequent minor estimation error. Nevertheless, the downside of these methods is that they typically necessitate a substantial amount of data regarding the degradation trend of training batteries. The most effective means of acquiring such data involves specifically designed degradation tests aimed at measuring the discharge capacity of the battery over time, from the initial state to the failure threshold. However, the development of new battery degradation tests faces significant constraints. Building a battery degradation dataset requires testing a large number of batteries until the failure threshold is reached. Advancements in technology have considerably extended battery life, with failure conditions now achievable after over 1000 complete charge/discharge cycles. Generating such datasets requires high costs, and it demands extensive time, human resources, and specialized measurement equipment. Additionally, the diversity in battery types (varying in size, chemistry, anode and cathode specifications, etc.) and operating conditions (dependent on charge/discharge profiles, ambient temperature, etc.) significantly impacts battery responses during degradation tests, leading to the necessity of developing numerous tests tailored to battery size, chemistry, charge and discharge current, temperature conditions, and more. Given these challenges, much of the recent literature typically evaluates the proposed RUL prediction method on degradation datasets publicly available. Several characteristics distinguish the most commonly used datasets for prognostic and RUL prediction [1,2,25]. In NASA’s [1] study, four 2000 mAh lithium-ion batteries were tested at standard operating conditions until 70% of rated capacity was reached (i.e., 24 °C ambient temperature, 1 C discharge profile, and 0.75 C constant current–constant voltage charge profile). Toyota [2] presents data from 140 lithium-ion phosphate/graphite cells with a nominal capacity of 1100 mAh. In the tests, the battery was charged and discharged as fast as possible at 30 °C until 80% of the nominal capacity was reached. According to the Center for Advance Life Cycle Engineering (CALCE) [25], prismatic cells with a capacity of 1100 mAh and 1350 mAh were tested at room temperature with a standard constant current–constant voltage (CC–CV) charging protocol at different C-rates until 80% of their nominal capacity had been reached. Detailed information about the testing protocols can be found in the references cited.

Stochastic and machine learning models have both advantages and disadvantages when used for RUL prediction. Stochastic methods are based on explicit probabilistic assumptions on the data-generating process. They have the advantage of an interpretable parameterization, allowing one to infer the relationship among the independent variables and the outcome, but it is required to establish assumptions consistent with the data-generating process in order to obtain accurate predictions. As such, stochastic methods are preferable when, in addition to prediction, analyzing the impact of independent variables is also of interest. Machine learning (ML) techniques are able to learn and recognize complex patterns based on experience, i.e., the aging patterns of batteries used in the training phase, that can be used for prediction and estimation. Most ML algorithms have the advantage of being able to handle non-linear patterns, which is an important ability when dealing with battery aging, leading to more scalable models, being applicable in different situations. The disadvantage of ML models is the need for huge amounts of data, which, as said before in this section, is time-demanding and expensive. To cope with this problem, ML models are usually accompanied by data augmentation techniques in order to create new synthetic data.

ML methods can be divided into classical models and deep learning (DL) models. Among classical ML models, the most used are support vector machine (SVM) and support vector regressor (SVR), such as in [26,27]. In contrast, when focusing on DL models, LSTM and Transformers are mainly used, due to their ability to deal with time series, and, with the use of the attention mechanism, to focus on the most relevant parts of the input, as in [28,29,30].

Another distinction that can be made when talking about ML models used for RUL estimation is the method used to obtain the RUL value. Most of the models use an indirect process to estimate the RUL, which is made in two, or in some cases three, phases. The other approach is to directly predict the RUL from the capacity fade curve in one step. Each approach has pros and cons: the indirect approach is more reliable when the capacity fade curve is noisier but, at the same time, is more time-consuming and computationally expensive; the direct approach is faster as it needs only one operation, but heavily relies on which data are available and how they are obtained. All the approaches to degradation modelling described above are developed and validated on single battery cells, either testing their own cells or taking common degradation datasets such as NASA, TOYOTA and CALCE as reference degradation trends for the cell. In contrast, studies of battery pack degradation are seldom available. Battery packs can experience unbalancing issues caused by the dependence between the cells, as well as the different intrinsic performances of each cell. Such unbalanced states can lead to complex degradation processes and abnormal discharge phenomena, which brings challenges in the SOH estimation of battery packs. For this reason, the study of battery pack degradation is still an open research question, made even bigger by the fact that publicly available datasets of battery pack degradation are not available. Despite the lack of public datasets available, some works in recent literature described and tested the degradation process of a battery pack enduring a life cycle assessment. For instance, [31] proposes a combination of transferred deep learning and Gaussian process regression, whereas an OCV-based method is presented in [32]. Battery pack degradation as a function of temperature is instead investigated in [33,34].

## 3. Specific Degradation Models

One of the purposes of this paper is to provide an overview of various methodologies documented in the literature for state-of-health (SOH) and remaining useful life (RUL) estimation and prediction. It is important to highlight that the literature presents different definitions of SOH. Therefore, building upon this observation, the paper reports the proposed SOH definition for each method, respecting the definitions presented in the original manuscripts. In this section, we want to describe in more detail some models to estimate battery degradation, which may be classified as data-driven since their assumptions do not take into account the physics of the degradation process. We will then show two ML models, one for each approach, direct and indirect, explained in the previous section, in order to obtain a global view of data-driven techniques.

### 3.1. General Path Models

General path models describe the degradation path through a parametric function of time and stress conditions. The most popular general path model is the linear model with random effects [7,35]. Consider the case of n batteries being tested for durability under constant load. A sequence of voltage measurements is provided for each battery, which is viewed as a sample unit randomly selected from a specified population of interest, such as that of a production lot. Battery degradation may be regarded as the sum of two components: a deterministic (unit-specific) function of time, for example a polynomial one, and a random (noise) component. In this case, the regression model can be formulated as follows:(1)$$Y_{i,t}=B_{i}^{\prime }x_{t}+\epsilon_{i,t}$$
where
$Y_{i,t}$ is the SoH of the *i*th battery measured at the measurement time t(t=1,...,T), which is assumed to be common across the units;$x_{t}={\left(1,t,t^{2},...,t^{p}\right)}^{\prime }$ where p is the degree of the polynomial time trend;$B_{i}={\left(B_{i,0},B_{i,1},...,B_{i,p}\right)}^{\prime }$ is the vector of length p+1 of the regression coefficients for the *i*th unit;$\epsilon_{i,t}$ is the random noise for *i*th unit at measurement time *t*.

The following distributional assumptions are made for the random terms in Model (Equation 1):$B_{i}\left(i=1,...,n\right)$ are random parameter vectors of length p+1 drawn from a multivariate normal distribution: $B_{i}∼MVN\left(\mu_{B},\Sigma_{B}\right)\left(i=1,...,n\right)$;noises $\epsilon_{i,t}\left(i=1,...,n\right)$ follow an autoregressive (AR) process of order q:$\epsilon_{i,t}=\sum_{k=1}^{q}\phi_{k}\epsilon_{i,t-k}+u_{i,t}$, where $u_{i,t}∼N\left(0,\sigma^{2}\right)$;$B_{i}$ and $\epsilon_{i,t}$ are independent of each other.

Model parameters can be estimated by maximizing the likelihood (maximum likelihood method) or, if a prior distribution incorporating quantitative knowledge can be assumed, by computing their posterior distribution (Bayesian approach). This second estimation method is useful when information on model parameters can be obtained from past studies. Then, given the failure threshold $y_{lim}$, which is the lowest acceptable level of the battery SoH, the predicted RUL of a battery of age τ is given by $x_{fail}-\tau$, where $x_{fail}$ is the time when the battery reaches $y_{lim}$, and is the numerical solution of:(2)$$Y\left(x_{t}\right)=B_{i}^{\prime }x_{t}=y_{lim}$$

This empirical random-effects regression model of polynomial type is sufficiently flexible to describe different behaviors of battery SoH observed during multiple simultaneous experimental tests. Splines can be used in place of simple polynomials to represent more irregular trends [7]. There are many parameters affecting battery degradation that are not taken into account in the proposed model, such as temperature, current density, pressure, fuel flow, and reactant concentration. Even when specific controllers are present, these parameters typically show little variation over time, even when they are kept constant for long periods of time. Therefore, since this model assumes steady-state conditions, the random noise ϵ exhibits an autocorrelated structure in each path. A degradation model that incorporates the above parameters (if available) might produce a lower autocorrelation. Furthermore, the above model could be improved by including other variables that influence the battery degradation process, such as temperature fluctuations, which affect (almost) all battery types simultaneously. In conclusion, it is noteworthy that the above model permits a separate quantitative evaluation of unit-to-unit heterogeneity and correlated noise variability.

### 3.2. Stochastic Processes

Whereas general path models assume that degradation has a deterministic trend, stochastic processes describe the evolution of degradation through a probability law on state transitions. Different stochastic processes have different characteristics; therefore, the choice of which one to adopt depends on the specific properties of the system under analysis. A stochastic process representing a degradation pattern consists of a collection of random variables indexed in time order: $W_{1},...,W_{t},...,W_{T}$. For example, using the Brownian motion process [10], it is assumed that increments in degradation are independent and normally distributed with null mean and variance equal to the temporal lag:$$W_{t+k}-W_{T}∼N\left(0,k\right)\;k\geq 0$$

The Weiner process is a generalization of the Brownian motion, where increments have a fixed non-null mean μ and their variance is scaled by a fixed positive factor $\delta^{2}$ [9]:$$W_{t+k}-W_{T}∼N\left(\mu ,\delta^{2}k\right)\;k\geq 0$$

Another stochastic process that has been used to model degradation phenomena is the autoregressive integrated moving average (ARIMA) process [8], which assumes that the current degradation level is a function of past degradation levels and past measurement errors. An ARIMA process of order (p,d,q) is defined as:$$\Delta^{d}W_{t}=\phi_{0}+\sum_{j=1}^{p}\phi_{j}\Delta^{d}W_{t-j}+\sum_{j=0}^{q}\psi_{j}\varepsilon_{t-j}\;\epsilon_{t}∼N\left(0,\sigma^{2}\right)$$
where $\Delta^{d}$ denotes *d*-order differing, $\epsilon_{t}$ are realizations of a normal white noise process, $\psi_{0}=1$, and $\phi_{0},\phi_{1},...,\phi_{p},\psi_{1},...,\psi_{q}$ are real-valued parameters. Differently from the Wiener, Brownian, and ARIMA processes, the Gamma process has the attractive feature of monotonicity, which may be suitable for some degradation phenomena [11]. Using the Gamma process, increments in degradation are independent and follow a Gamma distribution with shape parameter equal to the temporal lag times a fixed factor α>0 and fixed scale parameter β>0:$$W_{t+k}-W_{t}∼Gamma\left(\alpha k,\beta \right)\;k\geq 0$$

Recently, the inverse Gaussian process has been also used to model degradation phenomena [12]. In this case, increments in degradation follow an inverse Gaussian distribution.

### 3.3. Exponential Models

Lithium-ion batteries exhibit cumulative degradation and progressive wear-out, which are the primary contributors to the decline of the battery life. Such wear-out usually leads to:An exponential decrease of the battery’s discharge capacitance over the battery’s operational lifespan;An exponential rise in the equivalent series resistance (ESR) over the battery’s operational lifespan.

The exponential degradation of discharge capacity and internal resistance is shown in Figure 3, taking as an example five batteries from the Toyota dataset [2] that are randomly selected. For this reason, literature widely discusses describing these phenomena through the application of exponential degradation models. A commonly employed model within this category, as found in numerous recent studies, is the double exponential model [14,36,37,38,39]. The double exponential model strikes a favorable balance between accuracy and complexity, making it a good candidate for an accurate description of the battery degradation. As a matter of fact, examples of application of the double exponential model can be found considering the NASA dataset in [40], the CALCE dataset in [41], and the Toyota dataset in [42]. The most common form of the double exponential degradation model available in literature relies on four parameters, with two (i.e., *a* and *c*) representing the internal impedance of the battery, and the other two (i.e., *b* and *d*) denoting the aging rate. In essence, the state of health is a *k*-dependent function (where *k* is the number of cycles endured by the battery and it stands for a time variable) expressed as follows:(3)$$SoH_{k}=a\cdot e^{bk}+c\cdot e^{dk}$$

As pointed out by multiple studies (see reference above), the double exponential model is a rapid, accurate, and efficient way to estimate the discharge capacity of the battery and to forecast its degradation process. However, in some cases, other exponential degradation models can be used to improve the forecasting accuracy, allowing for a more precise RUL estimation and an optimal maintenance management. A valid alternative was published in [23]. In this case, a single exponential degradation model was assumed in order to describe and predict the degradation trend of the battery’s discharge capacity. More specifically, the discharge capacity $C_{k}$ at the *k*-cycle depends only on two parameters, one of them (i.e., *a*) linked to the internal impedance of the cell, and the other one used to model the aging rate (i.e., *b*). Thus, the single exponential degradation model able to describe the discharge capacity of the cell is expressed as follows:(4)$$C_{k}=C_{0}+a\cdot e^{b/k}$$

As can be noted in (Equation 4), the single exponential model requires the estimation of only two parameters. However, it is necessary to introduce an additive constant $C_{0}$, which is different for every battery, and it represents the initial capacity of the cell before its first working cycle. It is important to note that this constant is different from the rated capacity expressed by the manufacturer in the data sheet of the product, and thus it must be measured by the user with a dedicated procedure before the actual installation on the field. In order to estimate the real initial capacity $C_{0}$ of a battery, it is necessary to fully charge the cell before fully discharging it. An adequate resting time must be observed to ensure trustworthy results due to the presence of stabilization processes, thermal dissipation, voltage recovery, and hysteresis issues. The charge phase of this procedure must be carried out using a constant-current constant-voltage (CC-CV) profile at low rate (i.e., low charge current) to ensure that the 100% SOC condition has been reached. The discharge phase must be carried out using a constant current profile at a discharge rate similar to the condition that the battery will endure in the actual installation in the field. The initial capacity $C_{0}$ is then obtained by integrating the current measured during the entire discharge cycle over the discharge time interval. If the battery’s health indicator is defined as the ratio between the current capacity $C_{k}$ and the rated capacity $C_{rated}$, then when using the single exponential degradation model in (Equation 4), it is possible to forecast the degradation of the SOH as follows:(5)$$SoH_{k}=\frac{C_{k}}{C_{rated}}$$

The proposed single exponential model has been tested on two publicly available datasets, the NASA dataset in [23] and the Toyota dataset [4]. In both cases, the goodness of fit of the single exponential model has been proven to be comparable to the double exponential degradation model. However, the most interesting finding is related to the extremely better performances that the single exponential model has in forecasting the future degradation and the future SOH with respect to the double exponential model. This led to significantly better performances in the estimation of the RUL. The latter is the time difference between the current battery’s cycle and the future moment in which the battery is supposed to reach the failure threshold. According to the application field, the failure threshold can be set as the 70% or 80% of the rated capacity (i.e., when the SOH reaches 0.7 or 0.8). Considering both failure thresholds, authors have proven the ability of the single exponential degradation model to outperform the double exponential model in [4,23]. Finally, it is important to mention that both single and double exponential degradation models can be used to accurately predict the RUL of a battery following different approaches:Using it as fitting model in a curve fitting toolbox. This is the most easy and least complex algorithm but, at the same time, it is the less accurate.Using it as state space of a Kalman filter or particle filter. This is the most common way found in literature for the double exponential model.Using it to train a machine learning (ML) algorithm like regression models and support vector machine.Using it to train a deep learning algorithm. The case of a recurrent neural network was investigated and tested in [23], pointing out better performances than the classical filter algorithms and the ML algorithm.

The main advantage of both exponential models described above is the ability to correctly characterize the battery degradation in light of the extremely high accuracy of fitting the battery discharge capacity. However, this is also their intrinsic major limitation, since they need a continuous measurement of the discharge capacity of the battery during its entire operating life. Such measurement requires current transducers and online continuous monitoring of the battery, which is not always reasonable because of cost, dimension, and resources constraints.

### 3.4. Polynomial Model

It is typically assumed that battery aging is a function of the number of cycles it has been put through. Reference cycles are necessary for the definition and use of a health index based on them. Nevertheless, the authors have shown in [43,44] how capacity fading can be related to cumulative charge moved into the battery, regardless of the shape of the cycle (this property requires a constant temperature and current and a 20% to 80% charge state). It is possible to use the moved charge as a good indicator of battery health and as an unambiguous battery usage indicator. Literature presents various models analyzing the relationship between cell capacity *C* and moved charge *q* based on this consideration. An aging curve model using a polynomial function of third degree was developed by [43]:(6)$$C\left(q\right)=b_{0}+b_{1}q+b_{2}q^{2}+b_{3}q^{3}$$

Based on experimental findings, aging curves become almost linear when the residual capacity is under 95%, and a modified irrational function is applied to account for both linear and square terms [44,45]:(7)$$C\left(q\right)=b_{0}+b_{1}\sqrt{q}+b_{2}q+b_{3}q^{2}$$

Finally, letting $SoH\left(q\right)=C\left(q\right)/C_{in}$, where $C_{in}$ is the initial capacity of the battery, and dividing (Equation 7) by $C_{in}$, we obtain a degradation model for the SoH. In this scenario, SoH will be measured on a “moved charge” scale (i.e., $q_{fail}-q_{now}$) rather than a time scale (such as calendar time or number of cycles), where q fail is the solution to SoH(q)=80%, and $q_{now}$ is the current value of the cumulative moved charge.

### 3.5. Transformer Model

To explain the indirect RUL estimation process, we focus on the Transformer model. In recent years Trasformers became widely used for tasks linked to time series forecasting, and so also to RUL estimation. They can be considered an evolution of recurrent neural networks (RNNs) since they overcome many of their disadvantages. Transformers have the ability to model long-term dependencies and are also computationally more efficient. Furthermore, Transformers make use of the self-attention mechanism, which allows them to give different weights to each sample of the input, in this case, the capacity fade curve, in order to focus on the most important ones. When dealing with RUL estimation, Transformers cannot be used to directly regress the RUL, since the next expected output is strictly linked to the previous one, in particular $y_{t}=y_{t-1}-1$, and the model could learn only this relation, propagating a previous error in the prediction. Therefore, in order to obtain an indirect RUL estimation, we need two steps, with an optional one at the beginning:**Denoising** (optional): data are denoised using methods such as wavelet denoising, or a denoising auto-encoder [30];**Capacity forecasting**: the capacity fade curve is forecasted until it reaches 80% of its original value;**RUL estimation**: in the final step the RUL is estimated as the number of forecasts made in the previous step before reaching the EoL threshold.

The input of the model could be the full capacity fade curve of the battery under study, starting from the nominal capacity until the last measured value, or only a sliding window of the curve, containing the last *n* capacity values.

### 3.6. Conv-LSTM with Attention Mechanism

We conclude this section with the analysis of a DL model used for direct RUL estimation. LSTM is a type of RNN used for processing sequential data that addresses the vanishing gradient problem, which can occur when trying to learn long-term dependencies in sequential data. The LSTM architecture uses a memory cell that can store information over time and selectively update or forget it based on the current input. In this case, the model is composed of four main elements:**Convolutional layers**: whose purpose is to reduce the dimensionality of the input and at the same time maintain the information contained in it;**LSTM layers**: to obtain temporal information contained in the data;**Attention mechanism**: similar to the self-attention used in Transformers, adds a weight to each sample of the input;**Dense layers**: receive the weighted samples and produce the output.

Different from the indirect approach, the RUL is the output of the model, which avoids the forecasting of future capacity degradation. The input of the model is a sliding window of various data coming from battery aging tests, such as capacity degradation, temperature, and cycle number. The need for various input features is because this approach relies more on noise in the capacity fade curve, so using more features can deal with this problem.

## 4. Optimal Accelerated Testing and Maintenance Planning

For the design of accelerated degradation tests (ADT) and for planning maintenance strategies, the models presented in the previous section are essential. Our proposed approaches to both tasks are described in the following subsections, along with a brief review of optimal designs for ADT.

### 4.1. Optimal Designs for Accelerated Testing

Design of experiments is a broad and essential methodology of statistics theory, allowing one to assure data collection efficiency and suitable statistical modelling for estimation and prediction purposes. Appropriate planning and conducting of the experiment allow for obtaining highly informative data that ensure specific properties related to model estimation, inference, and/or prediction. In this context, an optimal planning of ADT enables one to find an efficient testing plan by considering one or more conditions imposed by engineers; for instance, requirement conditions related to the total number of test units, time and number of measurements, test stress levels, and similar. To this end, optimal designs are increasingly applied and studied in this context, where specific design optimality criteria allow for compliance with the final reliability aim(s). Optimal designs are model-dependent, and the design optimality is obtained based on the chosen design criterion and the underlying statistical model(s). In this regard, the general equivalence theorem provides the necessary and sufficient conditions for achieving and checking design optimality [46,47]. The wide range of design criteria available in the optimal design theory allows for obtaining optimal testing plans in the ADT field that align with the specific technological and reliability aim(s).

In the literature, optimal designs for ADT based on stochastic process models are extensively studied; more precisely, optimal designs for the Wiener process [48,49], the Gamma process [50,51], and the inverse Gaussian process models [52,53] are defined by considering both the constant stress loading and the step-stress loading [54]. Furthermore, in some recent studies dealing with general path models and optimal designs for ADT, primarily linear mixed-effects models are employed [55,56,57]. An important point to consider in the ADT field is that, usually, some further knowledge exists on the degradation process under study, for instance, available from the underlying product/process technology or from studies performed to study similar products and/or processes. To this end, Bayesian optimal designs are increasingly being studied and applied to obtain an efficient testing plan. More specifically, this is because they allow for accounting for the available prior information on the process under study by considering the uncertainties related to the assumed statistical degradation models and/or model parameters.

As previously described, when considering optimal designs, the design optimality is obtained based on the chosen design criterion. In this regard, when considering the Bayesian design framework, it must be noted that the design criterion is a function of the posterior distribution and is defined in terms of a utility function [58,59]. Therefore, to obtain the final optimal design for the test plan, the expected utility function should be optimized over the design space with respect to the future data (that is, the test outcome) and model parameters. Thus, the utility function is a key point, and it could be defined by considering the specific technological and reliability aim(s) and also issues related to costs; for instance, we may consider the Kullback–Leibler divergence in defining it [60,61,62], as well as utility functions for prediction of future observations [58].

It must be noted that few contributions are currently available in the literature dealing with Bayesian optimal designs for ADT. For instance, Shi and Meeker (2011) develop Bayesian optimal designs for accelerated destructive degradation tests based on a class of nonlinear degradation models and by considering only one accelerating variable [63]. Assuming a drift Brownian motion in the degradation model, Li et al. (2015) deal with Bayesian optimal designs for step-stress ADT [64]. Furthermore, Li et al. (2017) consider step-stress accelerated degradation testing and the inverse Gamma process to model the degradation path [65]; in this framework, they study Bayesian optimal design for ADT by considering the objectives of relative entropy, quadratic loss function, and D-optimality [65]. More recently, Weaver and Meeker (2021) [66] consider modelling the degradation path through linear mixed-effects models, and they develop a Bayesian criterion for the estimation of the quantile of the failure-time distribution for obtaining the optimal repeated measures ADT. Finally, it must be noted that, currently, sequential Bayesian optimal designs have gained considerable attention in the ADT field; more precisely, considering a dynamic programming framework for decision making, they allow for the obtaining of optimal maintenance policies. Currently, this line of research is an unexplored area, and only one recent study deals with some of these points in the context of structural reliability analysis [67]. Sequential Bayesian optimal designs appear especially well-suited for RUL prediction since they allow one to build design policies adaptively based on the current state, which depends upon earlier states. In this dynamic programming framework, each state contains suitable information to make decisions and describe the system’s evolution. It must be noted that two types of utility functions are defined and called rewards in this context. The first is the so-called stage reward, which depends on the observations, the design, and a given state. The second one is the terminal reward, which depends only on the final state. Therefore, the total utility function is obtained as the sum of the two types of rewards, the stage and the terminal one. To obtain the optimal Bayesian sequential design, the expected total utility function is optimized iteratively, taking into account the particular form of the defined transition function. To this end, appropriate utility functions are defined for the stage rewards and the terminal reward in accordance with the ultimate technological and reliability aim(s) and based on the characterization of the elements for establishing predictive maintenance strategies (states, the defined transition function, and statistical degradation model). The optimization presents a high challenge from a computational perspective, mainly due to the huge design space. Hence, employing and defining suitable algorithmic strategies, like simulation-based reinforcement learning, is crucial to iteratively enhance the expected total utility function while ensuring a feasible computational effort. In the ADT field, sequential Bayesian optimal designs are an unexplored area of research that should be further and extensively developed by considering several issues. More precisely, apart from suitable methods to deal with this computational challenge, further developments in this field may be related to the definition of the stage and terminal rewards related to the total utility function, as well as issues related to the inclusion of costs.

### 4.2. Maintenance Planning via Reinforcement Learning

There are different approaches to maintenance, according to the system type, the available data and the specific objectives of the maintenance activity. However, when real-time data on the system state are available, it is natural to look for strategies for condition-based or predictive maintenance, by leveraging up-to-date system information. In particular, reinforcement learning (RL) provides a useful framework to build such strategies, as recently proposed, for example, by [68] for condition-based maintenance, where the system changes state according to a degradation model and the selected maintenance action. With regard to predictive maintenance, [69] considers the optimization of a parameterized policy to choose inspection times, depending on the RUL and a system degradation model defined as a stochastic process with independent increments. It is possible to express this problem in the RL framework. Both [68,69] also include the estimation of the unknown degradation model parameters within the optimization procedure.

With regard to degradation models, the work in [70] provides a useful reference for the wide literature on this topic, in which the authors examine stochastic models from several classes. One common output of these models is that they provide the probability that the degradation level of the system crosses a set threshold in any given time interval, which is in a one-to-one correspondence with the probability distribution of the RUL. In this way, uncertainty about the future behavior of the system can be taken into account in maintenance planning.

Often, in condition-based and predictive maintenance, the choice of the maintenance action and the transition to the next degradation state depend only on the current degradation state, the maintenance decision, and additional events until the next decision time. This fact leads to the Markov decision process (MDP) framework that constitutes the basis of RL algorithms [71], which are aimed at considering the long-term consequences of actions to choose the best strategies and avoid ad hoc approaches. The MDP framework is suitable for maintenance strategies for lithium-ion batteries based on the SoH, because the SoH at the next decision time depends on the SoH at the current decision time and on the battery usage and environmental conditions across the decision interval. For every such interval, maintenance decisions can be limited to whether or not replace the battery or, depending on the available control actions, can also include how much energy to transfer to or from the battery. The RUL is the basic component of the value obtained from the operation of the battery, which, summed over all intervals until the SoH falls under a set threshold, will provide the cumulative reward of any specific strategy.

If the SoH is not directly observable, but only measured with an error, one has to resort to partially observable MDP (POMDP) and the associated RL algorithms. In this case, the degradation state is represented by a probability distribution called belief, which is updated according to the Bayes rule as new indirect measures become available [72].

Many algorithms to solve MDPs are available (see [71]), which can be exact or approximated, especially for large action or state spaces. In controlled conditions, if a degradation model of the battery is completely known, a solution may be feasible by dynamic programming or Monte Carlo RL, if we can compute expectations of the cumulative reward or at least obtain an approximation by simulating trajectories the degradation model. On the other hand, if usage and environmental conditions change dynamically, the increasing complexity of the state space requires different approximate approaches (such as temporal difference learning), possibly without a perfect knowledge of the additional dynamics in the system, including the behavior of the degradation model itself. In this case, a class of algorithms called Dyna agents is preferable, because it combines planning by simulation (using the current knowledge of the model) with learning of both the policy and of the model from actual observation of the environment and of the battery state.

In summary, we may state that existing contributions in RL methods for maintenance applications can also be useful for making decisions on battery operations and substitution, with different algorithms depending on what information is available: exact dynamic programming if the degradation model is perfectly known; RL combined with model learning if the latter is not perfectly known; solutions based on POMDP if the SoH of the battery is not directly observable.

## 5. Conclusions

In this paper, we describe how degradation models can be used in accelerated testing and maintenance planning for lithium-ion batteries. We have collected information about each topic from the literature to formulate an optimal accelerated test design and predictive maintenance plan. A key factor in decision making is the remaining useful life (RUL) of the battery. According to our research, RUL can be measured in different units, including charge/discharge cycles, calendar time, and moved charge. Calendar time must be taken into consideration when performing predictive maintenance operations in changing environments, and it must be matched to the battery’s lifetime unit. Under controlled experimental conditions, this problem can be avoided. Additionally, we highlighted the sequential nature of decision-making procedures, which suggests that degradation model learning should be conducted in conjunction with them.

## Figures and Tables

**Figure 1 sensors-24-03382-f001:**
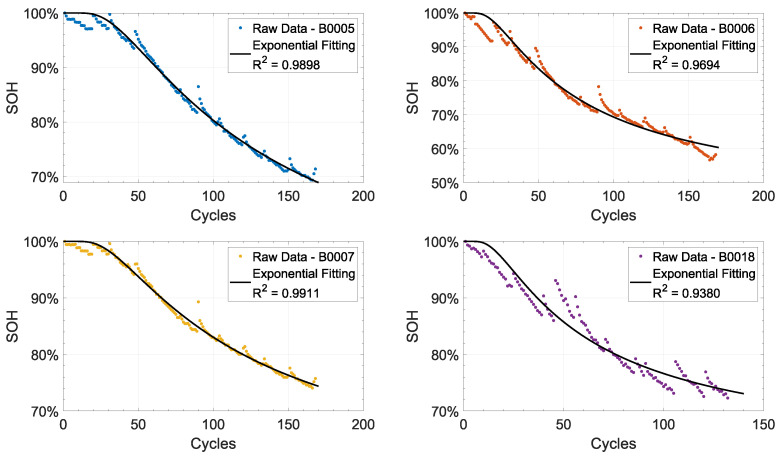
SoH of the batteries tested in the NASA dataset [1]. Each SoH has been fitted with a single exponential model. The $R^{2}$ value of each fitting is also reported.

**Figure 2 sensors-24-03382-f002:**
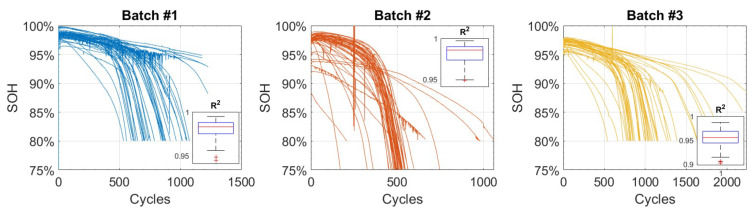
SoH of all batteries in the Toyota dataset [2] divided into three batches. Each subplot shows the $R^{2}$ value for an exponential model.

**Figure 3 sensors-24-03382-f003:**
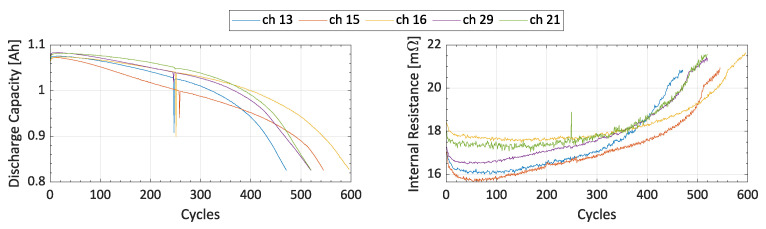
Discharge capacity and internal resistance of five batteries from the Toyota dataset.

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
