# Peer review of "A Review of Degradation Models and Remaining Useful Life Prediction for Testing Design and Predictive Maintenance of Lithium-Ion Batteries"

_sensors, 2024, doi:10.3390/s24113382_

Round 1

Reviewer 1 Report

Comments and Suggestions for Authors

Patrizi et.al., reviewed Li-ion battery degradation models and predictive maintenance. The  review may be accepted after minor revisions based on the suggestions below.

1. For RUL prediction cell data sets from NASA, Toyota, CALCE all are for cells and not packs, are there any data set source for packs? If not, how such analysis is done to identify fault in individual cells of a pack.

2. Limitations of each degradation models may be provided in detail.

3. Section on design of experiments for accelerated testing and predictive maintenance might require conclusive points listed.

Author Response

We would like to express our sincere gratitude to the reviewers for their meticulous work and for the valuable feedback that helped us improve our article.

*****************

Q0: Patrizi et.al., reviewed Li-ion battery degradation models and predictive maintenance. The  review may be accepted after minor revisions based on the suggestions below.

A0:  We thank the reviewer for the nice comments and interesting suggestions precious to improve the quality of the paper. We have carefully followed the proposed suggestions to further improve the quality of the paper.

Q1: For RUL prediction cell data sets from NASA, Toyota, CALCE all are for cells and not packs, are there any data set source for packs? If not, how such analysis is done to identify fault in individual cells of a pack.

A1:  Thank you for pointing out this interesting point of discussion. Battery packs can experience unbalancing issues caused by the dependence between the cells, as well as the different intrinsic performances of each cell. Such unbalanced state can lead to complex degradation processes and abnormal discharge phenomena, which brings challenges in the SOH estimation of battery packs. After a deep analysis of the literature review, to the authors knowledge, there is only one battery degradation dataset publicly available for battery packs (referenced here: https://doi.org/10.1038/s44172-023-00153-5) but it includes data about 1S2P configuration (i.e., a parallel configuration between 2 cells) which is still not enough for an exhaustive consideration about the degradation processes.  

Despite the lack of public datasets available, some works in recent literature described and tested the degradation process of a battery pack enduring a life cycle assessment. Generally speaking, all of these work showed a pseudo-exponential or pseudo-linear degradation trend of the battery pack, which is fully compatible with the trends obtained on a single cell by NASA, TOYOTA and CALCE. See for instance the following papers:

https://doi.org/10.1186/s10033-021-00668-y

https://doi.org/10.1016/j.est.2023.107270

https://doi.org/10.1016/j.est.2022.105216

To address the reviewer concern, we have updated the manuscript clarifying this issue in the Section 2 and explaining why only individual cells are considered in this work.

Q2: Limitations of each degradation models may be provided in detail.

A2:  We would like to thank the reviewer for the useful suggestion that helped us improve the quality of the manuscript. To fully address the Reviewer’s concern, we have updated the manuscript Section 3 including limitations and drawbacks of each degradation model studied and reported in our manuscript.

Q3:  Section on design of experiments for accelerated testing and predictive maintenance might require conclusive points listed.

A3: We thank the Referee for pointing out this issue. At the end of Sections 4.1 related to the design of experiments for accelerated testing, and at the end of Section 4.2 we include two new sentences as conclusive points.

Reviewer 2 Report

Comments and Suggestions for Authors

The current manuscript shows a significant degree of similarity with the article published by the same authors at the 2023 IEEE International Conference https://doi.org/10.1109/MetroXRAINE58569.2023.10405766. The attached similarity report highlights almost 100% similarity in the abstract, introduction, and conclusions sections between the current submission and the previously published work. This raises concerns about the originality and novelty of the current review article. Therefore, I must regretfully decline publication of this article.

Author Response

Thank you for bringing this matter to our attention. We would like to highlight that while the core structure of the manuscript remains the same, significant modifications have been made to the content, including the incorporation of updated research findings, expanded discussions, and additional references. The body of the text has been substantially revised to provide a more comprehensive review of the topic. We d are open to further revisions as necessary to address any concerns.

Reviewer 3 Report

Comments and Suggestions for Authors

I would like to thank the authors for the extensive literature review presented in this manuscript.

Sadly, I found very serious issues with the presentation of the work:

1. The main contribution is the compilation and presentation of the literature on models for RUL estimation, design of accelerated ageing tests and maintenance planning. This is not made clear by the title and introduction of the manuscript. I suggest the authors to modify the title and introduction in order to make the objectives of the review clearer to the reader.

2. As a literature review paper, I find the discussion of the reviewed works in sections 3 and 4 very limited. For example, in section 3, models for RUL estimation are presented, but at no point are the presented approaches compared and no discussion regarding when to favor an approach over the others is included. The same applies for section 4.

3. In section 3 the discussion of the multiple types of RUL estimation models, needs to be homogenized. For example, the definition of SoH is different for every type of model, making difficult comparing approaches.

4. Section 4 is very difficult to follow, maybe it can be improved be including tables that summarize and compare the discussed literature for each topic.

5. I suggest to extend the discussion presented in section 5, as currently it is very superficial.

Comments on the Quality of English Language

I found multiple grammatical and typing errors. I suggest to perform a full revision of the manuscript. Also, in some places I found the use of the term battery capacitance, instead of capacity.

Author Response

We would like to express our sincere gratitude to the reviewers for their meticulous work and for the valuable feedback that helped us improve our article.

*****************

Q0: I would like to thank the authors for the extensive literature review presented in this manuscript.

Sadly, I found very serious issues with the presentation of the work:

A0:  We thank the reviewer for the careful review and the useful suggestions that help us to improve the quality of the paper. We have carefully followed every single advice to further enhance the clarity and the quality of the manuscript.

Q1: The main contribution is the compilation and presentation of the literature on models for RUL estimation, design of accelerated ageing tests and maintenance planning. This is not made clear by the title and introduction of the manuscript. I suggest the authors to modify the title and introduction in order to make the objectives of the review clearer to the reader.

A1:  Thank you for pointing out this issue in our manuscript. According to your comment and the comments of other reviewers, we have updated the manuscript substantially revising both abstract and introduction sections to make clearer the aim and contribution of this work. Furthermore, we will ask the permisison to update the manuscript’s title to further address your concern. 

Q2: As a literature review paper, I find the discussion of the reviewed works in sections 3 and 4 very limited. For example, in section 3, models for RUL estimation are presented, but at no point are the presented approaches compared and no discussion regarding when to favor an approach over the others is included. The same applies for section 4.

A2:   Thank you for your suggestions. In the revised manuscript, we have added more references to extend our review, and we have also updated most of sections 3 and 4 to further improved and deepened the discussion of the methods.  

Q3:  In section 3 the discussion of the multiple types of RUL estimation models, needs to be homogenized. For example, the definition of SoH is different for every type of model, making difficult comparing approaches.

A3:  This is an interesting point of view since SOH definition of batteries (as well as other equipment) is not uniquely defined. However, this paper tried to present multiple approaches available in literature for SOH and RUL estimation and prediction. We appreciate your suggestion, and we discussed a lot within our team, however we believe that to fully discuss the main features and performances of each method it is more suitable to maintain the SOH definition of the original manuscript. Thus, we have decided to maintain different SOH definition for each method, according to their definition in the original manuscript. To fully address the reviewer concern, we have updated the manuscript clarifying this issue.

Q4:  Section 4 is very difficult to follow, maybe it can be improved be including tables that summarize and compare the discussed literature for each topic.

A4:  Thank you for this interesting suggestions. In the revised manuscript, we have completely rewritten section 4 including more conclusion and take-home messages for the main part of the section.  

Q5:  I suggest to extend the discussion presented in section 5, as currently it is very superficial.

A5:  Thank you for this comment. In the revised manuscript, we have completely rewritten section 5 discussing more in detail the aim and results of this work.  

Reviewer 4 Report

Comments and Suggestions for Authors

Review of the manuscript "A Decision-Making Approach to Lithium-Ion Batteries Degradation Modelling for Predictive Maintenance"

The authors propose a review of degradation modelling for accelerated battery degradation tests. Certainly, there are quite a lot of literature about degradation and batteries, although a good review is always interesting in any field, plus, if it is carried out by qualified researchers.

Thus, I can suggest few ideas to improve the quality of the manuscript:

- One method to quantify degradation is through SoH. Please, clearly and carefully define it. In line 27 you say that SoH is inferred from either Voc or Rseries. This is strange to me, as the most common way to define it is with the ratio between actual capacity and nominal capacity, which is in fact, stated below. Yet, Voc in not a typical way to measure SoH but SoC. And with respect to resistance, it is considered that the degradation threshold is reached when it has doubled in fatigue or wear processes. Also, fig 1 refers to SoH in %, which is usually related to capacity fade. See equation 5

-In line 32, with respect to fig 1, you say that an exponential decay can be fitted. Probably is true, but it looks like a linear dependence. Fig 2 migth ressemble more an exponential.

- You differenciate between physical models and data-driven models, with special emphasis in the latests. This certainly is the general trend. However, you could develope a little bit more the phyisical degradation models for completitude. For instance, Unified Mechanics theory proposed by Basaran could fit this approach. Or LLI and LAM decay described by Bor Yan Liaw, for instance.

- You claim that because it is difficult to know the precise physical models, data driven models are more convenient. Well, in bayesian models for instance, you rely on the prior information and the likelihood function. This likelihood is strongly related to the battery performance (in other words, physical mechanisms). At least, you can point out this issue. Otherwise, it seems that you can infer causal degradation relationships out of the blue from random data.

- lines 96 to 101 I found these lines quite vague. Could you be more precise in clarifying what you want to say? in fact, many passages are rather vague. Maybe the introduction of schemes or brief mathematical description of each approach could be helpful.

- Section 3.2 has is an expansion of lines 63-66 without any new information. I would suggest to rewrite or remove.

- line 217. Could you please provide a reference that the increase of ESR is exponential?. this is very surprising to me. I have always seen linear growth until failure.

- Paragraph starting in 231 starts saying that a common approach is to fit data with two exponentials and the authors claim that one exponential is enough. Being that true, please, rewrite in order to make it more logical and coherent. Why previous researchers have preferred the complexity of two exp if one was enough?

- line 250 adequate timing is very vague. What do you mean?

- Line 251 hystheresis has nothing to do with time and relaxation.

- line 257 describe better the end of discharge. A constant current discharge down to the threshold voltage does not reach the end of capacity, as you can expect the remnant capacity.

-ADT are good methods for lab characteritasition but are way far from real battery performance. Thus, to be able to predict battery behavior, you should include a passage describing the deviation of real battery operation to lab operation.

- Prognosis for maintenance rely on quantifying probabilities. The review is not very deep in describing how to quantify it. Could be a possible way for a better prognosis understanding.

Author Response

We would like to express our sincere gratitude to the reviewers for their meticulous work and for the valuable feedback that helped us improve our paper.

***********************************

Q0: Review of the manuscript "A Decision-Making Approach to Lithium-Ion Batteries Degradation Modelling for Predictive Maintenance"

The authors propose a review of degradation modelling for accelerated battery degradation tests. Certainly, there are quite a lot of literature about degradation and batteries, although a good review is always interesting in any field, plus, if it is carried out by qualified researchers.

Thus, I can suggest few ideas to improve the quality of the manuscript:.

A0:  We thank the reviewer for the careful review and the useful suggestions that help us to improve the quality of the paper. We have carefully followed every single advice to further enhance the clarity and the quality of the manuscript.

Q1: One method to quantify degradation is through SoH. Please, clearly and carefully define it. In line 27 you say that SoH is inferred from either Voc or Rseries. This is strange to me, as the most common way to define it is with the ratio between actual capacity and nominal capacity, which is in fact, stated below. Yet, Voc in not a typical way to measure SoH but SoC. And with respect to resistance, it is considered that the degradation threshold is reached when it has doubled in fatigue or wear processes. Also, fig 1 refers to SoH in %, which is usually related to capacity fade. See equation 5.

A1:  We have rewritten this section in order to make it more clear.

Q2: In line 32, with respect to fig 1, you say that an exponential decay can be fitted. Probably is true, but it looks like a linear dependence. Fig 2 migth ressemble more an exponential.

A2:   Thank you for your suggestion. We have modified fig. 1 in order to put in evidence the exponential decay.

Q3:  You differenciate between physical models and data-driven models, with special emphasis in the latests. This certainly is the general trend. However, you could develope a little bit more the phyisical degradation models for completitude. For instance, Unified Mechanics theory proposed by Basaran could fit this approach. Or LLI and LAM decay described by Bor Yan Liaw, for instance.

A3: 

Thank you for your feedback and for highlighting the importance of physical degradation models. While we have primarily focused on data-driven models in our review, we acknowledge the significance of including discussions on physical degradation models for completeness. However, in this particular manuscript, we have opted not to introduce additional analyses beyond the scope of our current discussion. We believe that maintaining our focus on data-driven models aligns with the objectives and scope of this review article. Nonetheless, we appreciate your suggestion and will keep it in mind for future research or publications. Thank you for your understanding.

Q4:  You claim that because it is difficult to know the precise physical models, data driven models are more convenient. Well, in bayesian models for instance, you rely on the prior information and the likelihood function. This likelihood is strongly related to the battery performance (in other words, physical mechanisms). At least, you can point out this issue. Otherwise, it seems that you can infer causal degradation relationships out of the blue from random data.

A4:  Thank you for your suggestion. The paper has been deeply revised, I hope that we have clarified this point.

Q5:  lines 96 to 101 I found these lines quite vague. Could you be more precise in clarifying what you want to say? in fact, many passages are rather vague. Maybe the introduction of schemes or brief mathematical description of each approach could be helpful.

A5:  thank you for your suggestion. We have revised the text.

Q6:  Section 3.2 has is an expansion of lines 63-66 without any new information. I would suggest to rewrite or remove.

A6:  We agree with your consideration and for this reason we have introduced a new section

Q7:  line 217. Could you please provide a reference that the increase of ESR is exponential?. this is very surprising to me. I have always seen linear growth until failure.

A7:  Thank you for this interesting comment. We understand that this concept could actually be counterintuitive. However, analyzing the data provided in the TOYOTA dataset about 140 batteries cycled until reaching the end of life, it is possible to appreciate that ESR actually follows an exponential growth when battery ages, while a linear fitting model is not always correct. To address the reviewer concern, we have updated the manuscript adding the required reference.

Q8:  Paragraph starting in 231 starts saying that a common approach is to fit data with two exponentials and the authors claim that one exponential is enough. Being that true, please, rewrite in order to make it more logical and coherent. Why previous researchers have preferred the complexity of two exp if one was enough?

A8: Thank you for pointing out this issue. A double exponential degradation model is the common way to describe degradation mechanisms of batteries according to several studies. This type of degradation model is suitable for data-driven approaches based on Artificial Intelligence or Kalman filter. However, a recent study (published by one of the authors of this work) discovered that a single exponential model fit extremely well the battery degradation processes of NASA and TOYOTA datasets, reaching similar or even better performances of the widely known double exponential. The comparison between the single and double exponential models is reported in:

10.1109/TIM.2021.3111009

10.1109/I2MTC48687.2022.9806707

To address the reviewer concern, we have updated the manuscript describing clearly such comparison in Section 3.3.

Q9:  line 250 adequate timing is very vague. What do you mean?

A9:  According to many manufacturer’s datasheet of different types of battery cells, adequate resting time between charge and discharge in life cycle test is between 20 and 30 minutes. To address your concern, we have updated the manuscript explaining in a clearer way the common meaning of adequate resting time.

Q10:  Line 251 hystheresis has nothing to do with time and relaxation.

A10:  Sorry for this inconvenience. We have updated the manuscript removing hysteresis reference in line 251.

Q11:  line 257 describe better the end of discharge. A constant current discharge down to the threshold voltage does not reach the end of capacity, as you can expect the remnant capacity.

A11:  Thank you for your indication. We have modified section 3.3, we hope this version is clearer.

Q12:  ADT are good methods for lab characteritasition but are way far from real battery performance. Thus, to be able to predict battery behavior, you should include a passage describing the deviation of real battery operation to lab operation.

A12: 

Thank you for your suggestion. We have rewritten this section, in order to explain how ADT can support laboratory involved in activities like battery behavior prediction

Q13:  Prognosis for maintenance rely on quantifying probabilities. The review is not very deep in describing how to quantify it. Could be a possible way for a better prognosis understanding.

A13: Thank you for your suggestion. Our purpose is to show how, for example, it is possible to define maintenance planning using a solution like reinforcement learning.

Round 2

Reviewer 1 Report

Comments and Suggestions for Authors

The manuscript may be accepted in its present form.

Reviewer 3 Report

Comments and Suggestions for Authors

I would like to thank the authors for addressing my comments. I find the current version suitable to be accepted. I have only a minor corcern regarding the abstract: I still believe that the main objectives of the article are not clearly presented. Otherwise, the paper was significantly improved.

      Comments on the Quality of English Language

I found some grammatical and typing errors. I suggest to perform a full revision of the manuscript. Also, in some places I found the use of the term battery capacitance, instead of capacity.

Reviewer 4 Report

Comments and Suggestions for Authors

The paper has included the improvements suggestes by the reviewers. It could be considered for publication.